

# Revealing the structure of precipitation extremes: a spatio-temporal wavelet approach

Svenja Szemkus, Sebastian Buschow, and Petra Friederichs

Institute of Geosciences, Section Meteorology, University of Bonn, Bonn, Germany

**Correspondence:** Svenja Szemkus (sszemkus@uni-bonn.de)

**Abstract.**

The impact of a heavy precipitation event is determined not only by the total amount of precipitation, but also by its spatial and temporal distribution. This study introduces a framework to quantify the key spatiotemporal properties of precipitation events - namely their characteristic time, length and speed - using radar-based observations. We employ a spectral filtering approach based on wavelet decomposition, which allows the selective extraction of precipitation signals at distinct temporal and spatial scales.

Focusing on Germany, we analyze the 100 most extreme two-day summer precipitation events using the high-resolution RadKlim dataset provided by the German Weather Service. We evaluate the physical plausibility of the derived characteristics and investigate their relationships with large-scale atmospheric dynamics. Our results reveal systematic patterns in the spatiotemporal organization of precipitation extremes. The framework presented here provides a robust tool for understanding extreme precipitation and offers potential for improved risk assessment and future climate studies.

## 1 Introduction

Heavy precipitation events repeatedly attract attention due to their catastrophic socioeconomic impacts. Prominent examples include the flooding events of July 2021 and May/June 2024 in Germany, as well as the severe flood episode in Emilia-Romagna, Italy, in May 2023 (e.g. Friederichs et al., 2024; Tradowsky et al., 2023; Arrighi et al., 2024). Taken together, these events demonstrate that extreme precipitation events pose a growing challenge for communities and policymakers and underscore the urgent need to better understand the underlying physical processes and possible changes in their characteristics in the context of ongoing climate change.

Major challenges remain in developing datasets that simultaneously provide sufficiently long time series and a temporal and spatial resolution fine enough to capture small-scale precipitation features. In practice, the effective resolution of a dataset is often lower than its nominal grid spacing, which can limit the representation of fine-scale processes. This applies, e.g., to products based on gridded station data, which typically have a much coarser effective resolution than their nominal spacing would suggest due to interpolation and smoothing effects. A similar limitation can also affect model-derived datasets if the underlying physical schemes do not explicitly resolve small-scale processes (see e.g. Ritzhaupt and Maraun, 2024; Pfahl et al.,





2017). Consequently, only high-quality radar observations and model products with sufficiently fine-scale physics can be considered reliable for analysing the spatio-temporal structure of precipitation events.

Moreover, precipitation manifests across a broad spectrum of spatial and temporal scales, ranging from short-lived, localized convective storms to long-lasting, synoptic-scale stratiform systems. The diverse physical processes governing these systems complicate efforts to robustly quantify changes in their frequency and intensity, as the inherent internal variability can mask underlying climate signals (see e.g. Fischer and Knutti, 2016).

In meteorology, it is common practice to describe the spatial-temporal organization of dynamical systems compactly using three primary metrics – characteristic length L, characteristic time T and characteristic speed U – which together determine the dynamic regime of a system. Building on this, a growing number of literature demonstrates a clear connection between large scale dynamics and the occurrence of extreme precipitation events (see e.g. De Vries, 2020; Kornhuber et al., 2019).

Traditional approaches to characterising the spatial and temporal structure of precipitation events often rely on fixed duration thresholds to examine different temporal scales and on counting the number of affected grid boxes to assess the spatial extent of precipitation events (e.g. Hundhausen et al., 2024; Voit and Heistermann, 2022; Lochbihler et al., 2017). Although such duration-based methods are intuitive and widely used, they suffer from an inherent redundancy: precipitation accumulations over longer periods inevitably include contributions from shorter accumulation periods. As a result, different duration classes do not provide independent representations of distinct precipitation-generating processes, but rather constitute nested aggregations of the same underlying signal.

In contrast, spectral filtering can separate temporal and spatial scales in an orthogonal and non-overlapping manner, allowing process-relevant variability to be isolated without redundancy. This approach not only facilitates the identification of distinct physical processes, but also enhances the detection of climate signals that may be obscured in the raw data. In this paper, we propose a spectral decomposition for the quantitative analysis of the spatio-temporal properties of precipitation events. Our method is based on wavelet transform (WT), which has been widely used across scientific disciplines for filtering different scale signals.

The WT is used because it's decomposition is based on localized basis functions. In contrast to, e.g., sinusoidal Fourier basis functions, wavelets are known to be particularly well suited for capturing both short-lived and long-lived, non-periodic features in the signal (see e.g. Mallat, 1989; Daubechies, 1992). Moreover, the WT has been intensively used in the analysis of spatial precipitation data, demonstrating its usefulness in various contexts, including the verification of spatial precipitation fields (see e.g. Casati et al., 2004; Yano and Jakubiak, 2016; Buschow and Friederichs, 2021), quantification of convective organisation (Brune et al., 2021), and the analysis of climate change signals (see e.g. Buschow, 2023; Benestad et al., 2022).

Wavelets in one, two and three dimensions, as well as corresponding algorithms, are available (see e.g. Selesnick and Li, 2003; Kingsbury, 1998); However, a common limitation is that filtering is typically performed using uniform scales across all dimensions. For example, a two-dimensional algorithm applies the same scale in both the x- and y-directions, while a three-dimensional algorithm treats the x-, y-, and z-directions equivalently. As a result, these approaches do not provide a straightforward or intuitive measure of combined spatial and temporal scales. Following an approach carried out by Yano and





Jakubiak (2016) in two dimensions, we combine a one-dimensional WT in time with a two-dimensional WT in space to achieve
a joint analysis of the spatial and temporal scales underlying precipitation events.

The focus of this study is on Germany, motivated by the availability of high-resolution, radar-based precipitation data. In
particular, we use the RadKlim dataset provided by the German Weather Service (DWD), which offers a continuous time series
of hourly precipitation data from 2001 to 2024 (Lengfeld et al., 2020).

Our analysis concentrates on the 100 most extreme precipitation events occurring during the northern hemispheric summer
months over Germany to investigate their spatiotemporal characteristics. Using a wavelet-based framework, we quantify the
dominant spatial and temporal scales of each event. From this, we derive the characteristic length, the characteristic time and
two independent descriptors - the characteristic scale and the characteristic speed - which provide a compact and physically
meaningful representation of precipitation events. We show that these parameters are closely related to atmospheric dynamics,
confirming the physical plausibility of our approach. By linking event-scale properties to larger-scale processes, this study
contributes to a deeper understanding of how precipitation extremes organize in space and time.

This paper is organized as follows. In Sect. 2, we present the theoretical background for the proposed space–time decompo-
sition approach. Section 3 describes the data and preprocessing steps in detail. In Sect. 4.1, we analyze selected representative
cases, followed by a comprehensive evaluation of the top 100 events in Sect. 4.2. The connection to the underlying dynamics
and thermodynamics is investigated in Sect. 5. Finally, Sect. 6 summarizes the main findings and provides a discussion of their
implications.

## 2   Theory

A variety of wavelet types - or wavelet families, exist, ranging in complexity from simple step-function wavelets (Haar, 1909)
to more advanced formulations. Each wavelet family is characterized by specific properties such as smoothness, symmetry,
and vanishing moments, which make it more or less suitable for capturing different features in a signal. For a comprehensive
introduction to wavelet theory, the interested reader is referred to standard references such as Daubechies (1992) and Mallat
(1998).

In this study, we employ the complex Dual-Tree wavelet which has proven effective for the analysis of spatial precipitation
fields (e.g., Buschow and Friederichs, 2021; Brune et al., 2021). The Dual-Tree Complex WT (DT-$\mathbb{C}$WT) and the algorithm
for its computation, were introduced by Kingsbury (1998). In principle, the proposed method can be implemented with any
wavelet family. We tested several alternatives, including Haar and Daubechies wavelets, but found no substantial performance
advantage over the DT-$\mathbb{C}$WT. The DT-$\mathbb{C}$WT, like other WTs, satisfies Parseval's energy theorem, an energy conservation
principle stating that the $L^2$-norm of the original signal equals the $L^2$-norm of its wavelet coefficients (see Kingsbury, 1998).
This ensures that the transform preserves the total energy of the precipitation field across scales.

We first introduce the one- and two-dimensional discrete WT in Sect. 2.1, before we extend to three dimensions in Sect. 2.2.
For consistency, we denote parameters associated with the one-dimensional WT with the subscript $_T$, and those associated with
the two-dimensional WT with the subscripts $_X$ and $_Y$ (or collectively $_{XY}$) throughout this section.





## 2.1 Discrete wavelet transform in one & two dimensions

For the one-dimensional WT, let $h(t)$ be a signal defined over discrete time steps $t_1, \ldots, t_{n_T}$ where $n_T$ is a power of two. Let $\psi(t)$ denote a wavelet function, with daughter wavelets $\psi^{k_T, j_T}(t)$ defined as:

$$\psi^{k_T, j_T}(t) = \frac{1}{\sqrt{2^{j_T}}} \psi\left(\frac{t - k_T 2^{j_T}}{2^{j_T}}\right). \tag{1}$$

$j_T$ and $k_T$ are discrete parameters controlling the scale and translation of the wavelet along the time axis, enabling the analysis of spectral energy at different temporal scales and positions. The wavelet coefficients $d(k_T, j_T)$ quantify the signal $h(t)$ at a given scale $j_T$ and temporal position $k_T$ and are defined as:

$$d^{j_T}(k_T) = \langle h(t), \psi^{k_T, j_T}(t) \rangle_t, \tag{2}$$

with $\langle \cdot, \cdot \rangle_t$ denoting the inner product in time.

To extend the wavelet analysis to two dimensions, we consider a signal $g(x, y)$, where $x$ in $x_1, \ldots x_{n_{XY}}$ and $y$ in $y_1, \ldots y_{n_{XY}}$ denote discrete spatial coordinates. We assume that $x$ and $y$ have the same length $n_{XY}$ and, analogous to the one-dimensional case, $n_{XY}$ is a power of two. The two-dimensional wavelets are scaled by a common scale parameter $j_{Y,X}$ and translated along the $x$- and $y$-axes by $k_X$ and $k_Y$, respectively, to cover different spatial positions. Directional information can be incorporated through an orientation parameter $q$ (see, e.g., Daubechies, 1992). This leads to the following definition of the daughter wavelets of the two-dimensional wavelet function $\psi(x, y)$:

$$\psi_q^{(k_X, k_Y), j_{X,Y}}(x, y) = \frac{1}{2^{j_{X,Y}}} \psi_q\left(\frac{x - k_X 2^{j_{X,Y}}}{2^{j_{X,Y}}}, \frac{y - k_Y 2^{j_{X,Y}}}{2^{j_{X,Y}}}\right), \tag{3}$$

and the two-dimensional wavelet coefficients:

$$d_q^{j_{X,Y}}(k_X, k_Y) = \langle g(x, y), \psi_q^{(k_X, k_Y), j_{X,Y}}(x, y) \rangle_{x, y}, \tag{4}$$

with $\langle \cdot, \cdot \rangle_{x,y}$ denoting the inner product in space. The wavelet coefficients $d_q^{j_{X,Y}}(k_X, k_Y)$ characterize the localized spectral properties of $g(x, y)$ across scales $j_{X,Y}$, locations $(k_X, k_Y)$ and orientations $q$.

In order to comply with Parseval's energy theorem, we summarize the spectral energy at each scale using the $L^2$-norm of the wavelet coefficients, which for the one- and two-dimensional cases is given by:

$$e^{j_T} = \sum_{k_T} |d^{j_T}(k_T)|^2, \tag{5}$$

$$e^{j_{X,Y}} = \sum_q \sum_{(k_X, k_Y)} |d_q^{j_{X,Y}}(k_X, k_Y)|^2. \tag{6}$$

Figure 1 illustrates the complex Dual-Tree wavelet in one and two dimensions. An important feature is that the real and imaginary parts are phase-shifted relative to each other, forming an approximately analytic and rotation invariant wavelet (Selesnick, 2002).





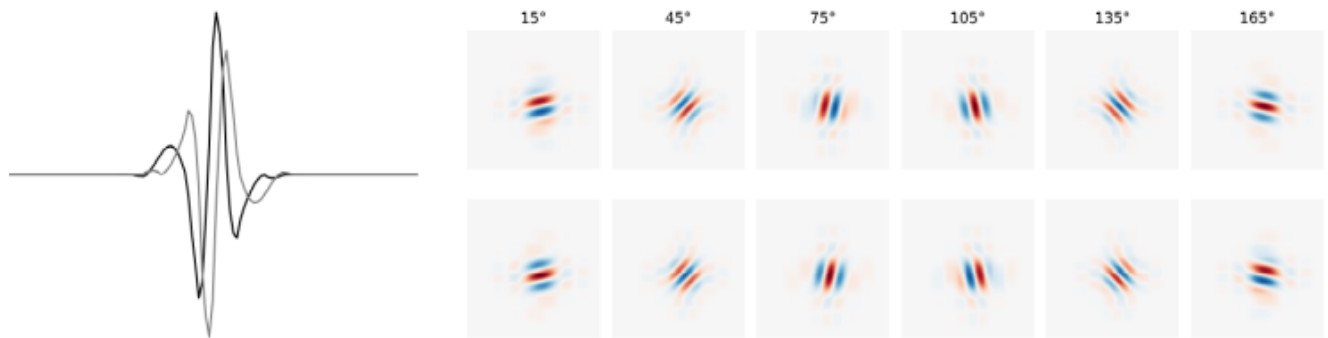

**Figure 1.** Illustration of the complex Dual-Tree wavelet in one (left) and two dimensions (right). In one dimension, the real and imaginary parts of the wavelet are shown by the black and gray lines, respectively. In two dimensions, the real and imaginary components are displayed in the top and bottom rows, with columns corresponding to different wavelet orientations ($15°$, $45°$, $75°$, $105°$, $135°$ and $165°$).

## 2.2 Discrete wavelet transform in spatio-temporal space

120 For the three-dimensional WT, we consider signal $f(x,y,t)$, encompassing spatial and temporal dimensions as pointed out in Sect. 2.1.

We then apply a one-dimensional WT along the temporal dimension $t$ and a two-dimensional WT for the spatial fields $(x,y)$ and calculate the wavelet coefficients as follows:

$$d_q^{j_{X,Y},j_T}(k_X,k_Y,k_T) = \langle\langle f(x,y,t),\psi^{k_T,j_T}(t)\rangle_t,\psi_q^{(k_X,k_Y),j_{X,Y}}(x,y)\rangle_{x,y}. \tag{7}$$

125 By doing so, we decouple the spatial and temporal scales, and provide individual scaling parameters for the spatial and temporal scales. This allows for an interpretable analysis of spatiotemporal characteristics within the signal $f(x,y,t)$.

Following Eq. 5 and Eq. 6, we summarize the spectral energies given by each combination of spatial and temporal scales as:

$$e^{j_{X,Y},j_T} = \sum_q \sum_{k_t} \sum_{k_x,k_y} |d_q^{j_{X,Y},j_T}(k_X,k_Y,k_T)|^2. \tag{8}$$

130 Our implementation is primarily based on the open-source Python package **DTCWT** (Wareham et al., 2013), which offers an implementations of the 1D, 2D and 3D DT-ℂWT based on the multiresolution approximation (MRA) algorithm (Mallat, 1989), which is typically applied for discrete wavelets. The restrictions on $n_T$ and $n_{XY}$ to be powers of two, as mentioned above are a direct consequence of the MRA algorithm. In a second consequence, we yield estimates of spectral energies only for the discrete spatial and temporal scales $j_T = 2^0, 2^1, \ldots, n_T$ and $j_{XY} = 2^0, 2^1, \ldots, n_{XY}$.





To derive the characteristic length $L = \bar{j}_{XY}$ and characteristic time $T = \bar{j}_T$ from the spectral energies $e^{j_{X,Y},j_T}$ (Eq. 8), we essentially follow Buschow (2022) and compute the center of mass as the average scales, weighted by the spectral energies:

$$\bar{j}_{XY} = \frac{\sum_{j_T} \sum_{j_{XY}} j_{XY}\, e^{j_{X,Y},j_T}}{\sum_{j_T, j_{XY}} e^{j_{X,Y},j_T}} \quad \text{and} \quad \bar{j}_T = \frac{\sum_{j_{XY}} \sum_{j_T} j_T\, e^{j_{X,Y},j_T}}{\sum_{j_T, j_{XY}} e^{j_{X,Y},j_T}}. \tag{9}$$

From these two characteristic properties, we derive the characteristic speed of the precipitation system as $V = L/T$. We further define a combined space-time scale, which we refer to as the characteristic scale $S$ as:

$$S = \sqrt{\left(\frac{L}{\text{std}(L)}\right)^2 + \left(\frac{T}{\text{std}(T)}\right)^2}. \tag{10}$$

By scaling $S$ and $T$ by their respective standard deviations, which we calculate across all 100 precipitation events considered in this study, we obtain dimensionless quantities that can be meaningfully combined despite originating from different units and value ranges.

Note that, analogously, the centre of mass can be derived from the one- and two-dimensional WT, in which case we obtain an estimate at each timestep for the characteristic length $L(t)$ and at each gridpoint for the characteristic time $T(x,y)$.

## 3 Data & Preprocessing

This study utilizes hourly precipitation totals from RadKlim, version 2017.002, provided by DWD (Winterrath et al., 2018). RadKlim is a high-resolution climatological dataset, provided on a $1\text{km} \times 1\text{km}$ grid which has been specifically developed to improve the understanding of precipitation dynamics and to support applications in climate research, hydrology, and risk assessment. The dataset combines radar-based precipitation estimates with observations from over $1,000$ ground-based gauges, using advanced correction algorithms to reduce radar-related biases. Temporal consistency is maintained through standardized processing and harmonized radar calibration, making the dataset suitable for long-term climate trend analyses. The record begins in 2001 and is updated annually, covering the period from 2001 to 2024 at the time of writing.

For our analysis in Sect. 5, we further apply data from the ERA5 reanalysis data (Hersbach et al., 2025). Specifically, we use 1) daily mean wind speed at the pressure levels 300, 500, 700, 850, and 1000 hPa, and 2) the convective time scale $\tau$ (Done et al., 2006). The latter is defined as the ratio between convective available potential energy (CAPE) and the rate at which CAPE is consumed by convection, thereby quantifying how rapidly potential energy is converted into convective motion. Small values of $\tau$ indicate stronger large-scale forcing, whereas large $\tau$ corresponds to airmass convection with only weak forcing from larger scales.

### 3.1 Event identification

From the available time period, we aim to identify the most intense precipitation events during the northern hemispheric summer months (June-August). Specifically, we select the top 100 events based on the total precipitation volume accumulated within a 2-day window over a spatial domain of $50\text{km} \times 50\text{km}$. While we acknowledge that this choice of temporal and spatial



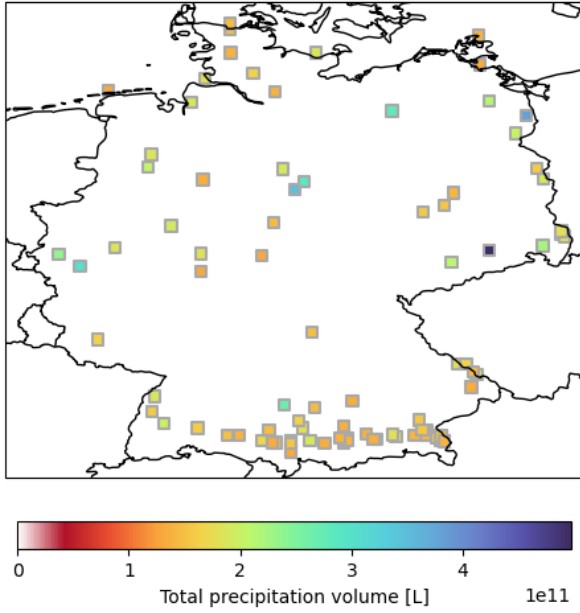

**Figure 2.** Center location and total precipitation amount [L] of the top 100 precipitation events in Germany, in terms of their 2d × 50km × 50km total precipitation volume (color) during the northern hemispheric summer months (June-August).

thresholds is to some extent arbitrary, we consider it a reasonable approximation for capturing high-impact events in which
large precipitation volumes fall over a limited area within a relatively short time. To avoid temporal clustering, events are
required to be separated by at least 72 hours. As a consistent, impact-oriented measure of event intensity throughout this paper,
we use the total precipitation volume within this region and time period for each event.

## 3.2    Data preprocessing

After identifying the events of interest, we analyze each event over a 4-day window centered on the precipitation peak (±48
hours) and across the entire spatial domain of RadKlim. This approach allows us to avoid manually defining the temporal or
spatial boundaries of each event, which would require arbitrary thresholds or other subjective criteria. While this may include
some precipitation not directly associated with the main event, its effect is considered negligible, as such additional rainfall is
generally weak and does not significantly alter the overall signal.

Extreme precipitation events are rarely confined to national or regional boundaries. Consequently, the RadKlim dataset
may not fully capture their complete spatial extent, a limitation commonly recognized in the analysis of large-scale hydro-
meteorological extremes. For instance, the 2021 Ahr flooding required combining Belgian and German precipitation datasets
to obtain a more comprehensive representation (Tradowsky et al., 2023). However, merging datasets introduces considerable
challenges, including increased processing effort and potential systematic inconsistencies. Therefore, in this study we restrict
our analysis to the spatial and temporal coverage provided by the RadKlim dataset.



We perform two additional preprocessing steps on the precipitation raw data. First, we ensure that the dimensions in time $(n_T)$ and space $(n_{XY})$ are a power of two. The RadKlim dataset provides a spatial resolution of $900 \times 1100$ grid points. To obtain a square domain suitable for further analysis, we zero-pad the x-dimension and remove the outermost rows along the y-dimension, resulting in a $1024 \times 1024 = 2^{10} \times 2^{10}$ grid. Temporally, we use a four-day interval comprising 94 time steps, which is extended to 128 time steps via zero-padding.

In a second preprocessing step, we smooth the dataset edges to prevent artifacts arising from sharp edges. While smoothing only the outermost grid points is usually sufficient, the dataset edges coincide with national and state borders. Therefore, we calculate the Euclidean distance from each grid point to the nearest border and apply a linear smoothing to zero within a 20 km margin. Additionally, the outer 10 time steps are linearly smoothed to zero.

## 4   Analysis of Extreme Precipitation Events

**4.1   Case studies**

We begin our analysis with two well known precipitation events over Germany: the August 2002 flood in Saxony and the July 2021 flood over the Ahr Valley, Western Germany. Both events rank among the top five identified in Sect. 3 and are considered among the most impactful precipitation events in German history (see e.g. DKKV, 2022).

     Figure 3 shows the characteristic length and characteristic time as derived from one- and two-dimensional WT. The one-
dimensional WT is computed at each grid point, while the two-dimensional WT is applied at each time step, providing spatially and temporally resolved estimates of the characteristic scales. Both characteristic length and time reach their maximum near the center of the event, with smaller-scale structures surrounding the main event in space and time. On average, the August 2002 event exhibits larger characteristic times (10.5 h) and lengths (82 km) compared to the July 2021 event (7.5 h and 74 km, respectively), indicating that the July 2021 event was, on average, more small-scale in nature.

These insights from one- and two-dimensional WT, while interesting, do not fully represent the structure of an event because time and space are treated separately. To overcome this, three-dimensional approaches are required, allowing for a joint analysis of spatial and temporal scales.

     Figure 4 shows the spectral energies $e^{j_{XY}, j_T}$ (Eq. 8) of the six most intense precipitation events, including the August 2002 and July 2021 cases. Each panel displays the spectral energy of the corresponding event, decomposed into 10 spatial scales
$(j_{XY})$ and 7 temporal scales $(j_T)$. The sum over all scales equals the total spectral energy, which corresponds to the variance of the precipitation field. This representation illustrates how the spectral energy of each event is distributed across discrete combinations of spatial and temporal scales. The patterns observed here are consistent with our findings from Fig. 3.

     Compared to the other events within the top six, the August 2002 event consistently shows a high proportion of energy at larger scales, together with the July 2017 event, which also displays substantial contributions at comparatively large scales -
reaching up to approximately 256 km and 34 h in 2002, and up to 500 km and 64 h in 2017.

     The events from 2010, and 2021 exhibit comparatively stronger energy contributions at smaller scales. These observations are consistent with previous studies highlighting the August 2002 event as a multi-day rainfall episode (e.g. Ulbrich et al.,





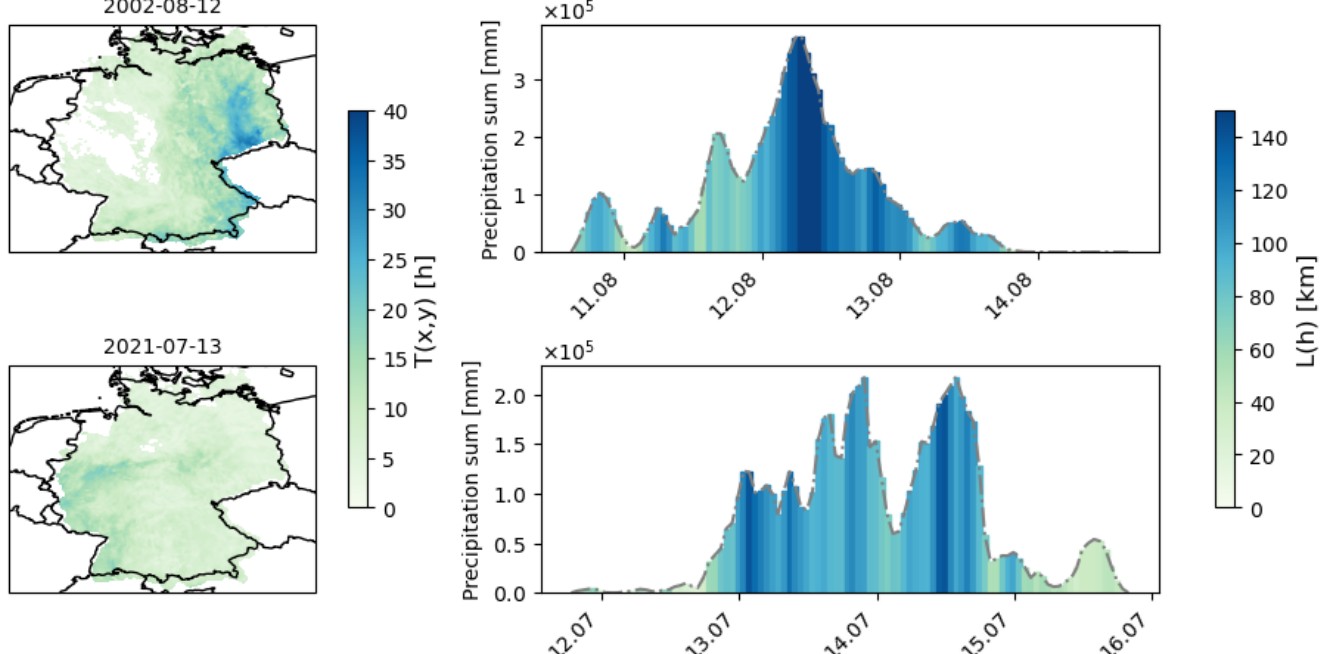

**Figure 3.** Characteristic time $T(x,y)$ (left) and precipitation sum with color indicating the characteristic length $L(t)$ (right). Shown are the two precipitation events August 2002 (upper row; based on data from 2002-08-10 16 UTC to 2002-08-14 17 UTC) and July 2021 (lower row; based on data from 2021-07-11 20 UTC to 2021-07-15 21 UTC). The precipitation sum is calculated as the total sum of precipitation amount over the respective time period and across Germany.

2003). By contrast, the July 2021 event was dominated by a series of very intense, short-duration convective rainfall events (e.g. Ludwig et al., 2023; Tradowsky et al., 2023).

A prominent and recurring feature across all spectra is the concentration of energy along a straight line with an approximate slope of one. The magnitude of the energy along this line varies between events, reflecting differences in the underlying space and time scales. Following Kraus (2004), the orientation along a straight line can be attributed to the movement of precipitation systems through space. A slope of 1 then corresponds to a constant velocity (no acceleration). Analogously, a slope of 2 would correspond to constant acceleration, which we do not further consider in this study. The value of the velocity can be inferred

from the y-intercept in this double logarithmic presentation. For reference, we included lines indicating a characteristic speed of 5km/h, 10km/h and 20km/h in Fig. 4.

   Precipitation events can exhibit distinct characteristic velocities across spatial and temporal scales. For instance, in the August 2002 event we find from Fig. 4 that small-scale features tend to propagate faster than large-scale components, as indicated by the orientation of spectral energies for different sized features, highlighting the complex, multiscale dynamics of

precipitation systems.



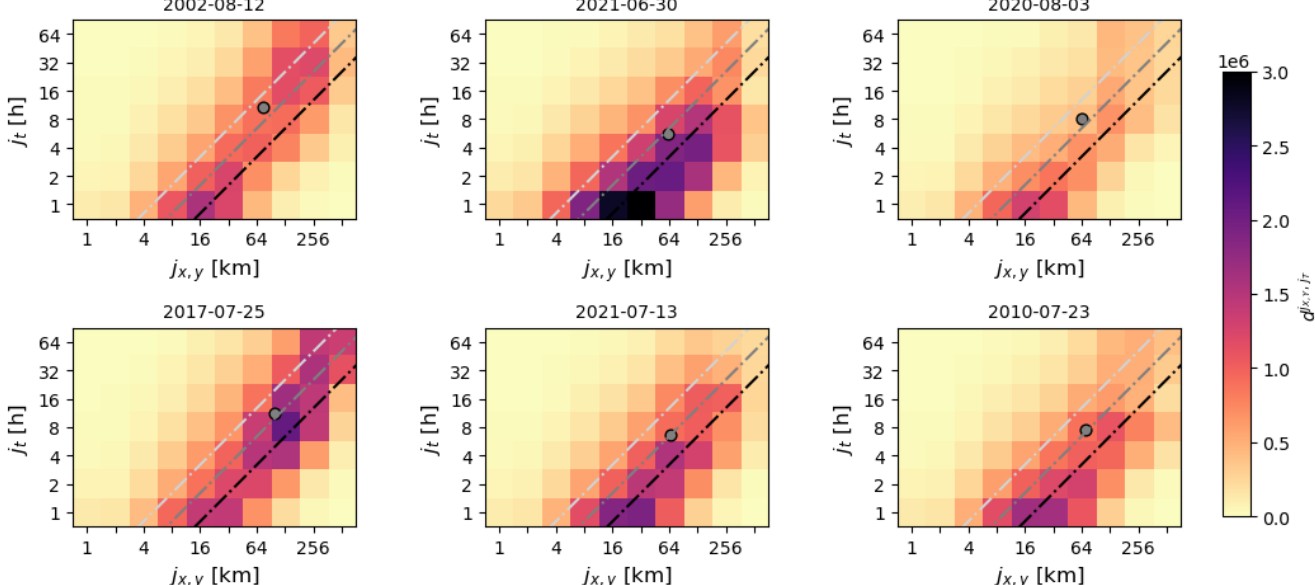

**Figure 4.** Wavelet spectrum (Eq. 8) of the top six precipitation events identified Sect. 3. The white, gray and lightgray solid lines denote a characteristic speed of 5km/h, 10km/h and 20km/h. The center of mass (Eg. 9) is indicated by a gray dot.

Finally, Tab. 2 summarises the characteristic properties of the top six precipitation events. For each event, we report the characteristic length $L$ and time $T$ (see Eq. 9), as well as the characteristic speed $V$ and scale $S$ (see Eq. 10). These metrics provide a concise representation of the spatial and temporal characteristics inferred from the spectral energy distributions in Fig. 4. For example, the comparatively large-scale nature of the August 2002 and July 2017 events is clearly reflected, and both are found to propagate relatively slowly. By contrast, the June 2021 event exhibits the smallest spatiotemporal scales and the highest propagation speed, followed by the flooding event in July 2021. The August 2020 and June 2010 events fall in between these two categories.

**Table 1.** Summary of mean spatiotemporal characteristics of the top six precipitation events identified Sect. 3. We report the characteristic length $L$, the characteristic time $T$ (Eq. 9), the characteristic speed $V$ and characteristic scale $S$ (Eq. 10) for each event.

|          | 2002-08-12 | 2021-06-30 | 2020-08-03 | 2017-07-25 | 2021-07-13 | 2010-07-23 |
|----------|-----------|-----------|-----------|-----------|-----------|-----------|
| $L$ [km]   | 106.37 | 88.82 | 92.15 | 137.08 | 93.23 | 100.84 |
| $T$ [h]    | 15.11 | 7.97 | 11.51 | 15.80 | 9.35 | 10.49 |
| $V$ [km/h] | 7.04 | 11.14 | 8.01 | 8.69 | 9.97 | 9.61 |
| $S$        | 8.43 | 5.58 | 6.78 | 9.65 | 6.13 | 6.75 |




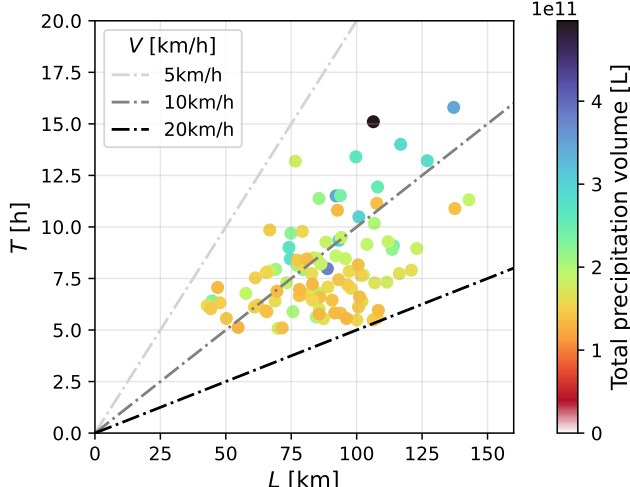

**Figure 5.** Scatter plot of the characteristic length $L$ and characteristic time $T$ (Eq. 9), for the top 100 precipitation events identified in Sect. 3. Point color indicates total precipitation volume similar to Fig. 2. Straight lines indicate a characteristic speed $V$ of 5 km/h (light gray), 10 km/h (gray) and 20 km/h (black).

## 4.2 Statistical Analysis

After concluding from Sect. 4.1 that the characteristic properties provide a suitable representation of the spatiotemporal struc-
ture of a precipitation event, we now extend the analysis to a stochastic assessment of the top 100 events using characteristic
length, time, speed and scale. Figure 5 summarises the characteristic length and characteristic time of the 100 most extreme
precipitation events identified in sec 3. The events span a wide range of scales, with characteristic lengths between 43 km and
143 km and characteristic durations between 5 and 16 hours. The corresponding propagation speeds range from 5 km/h to
20 km/h, which is physically plausible for precipitation systems. We further observe a tendency for high-volume precipitation
240  events to be more slowly propagating systems, which - consistent with physical intuition - would come with their enhanced
persistence. This feature will be examined in more detail in the following.

To further investigate the dominant spatiotemporal characteristics of the top 100 precipitation events, we apply a K-means
clustering algorithm to the spectral energies $e^{j_{XY}, j_T}$ (Eq. 8). To emphasize structural patterns rather than the absolute energies
of each event, the spectral energies are normalized individually as: $e^{j_{XY}, j_T} / \sum e^{j_{XY}, j_T}$. We assess the quality of the cluster
245  solution using the silhouette score (rou, 1987) and evaluate stability under resampling with the adjusted Rand index (Hubert
and Arabie, 1985) for K-means solutions ranging from 2 to 10 clusters (see Supplement Sect. A for full results). The highest
silhouette score (0.31) and robust stability (median ARI = 0.80) are achieved for a three-cluster solution, which we adopt for
the subsequent analysis.



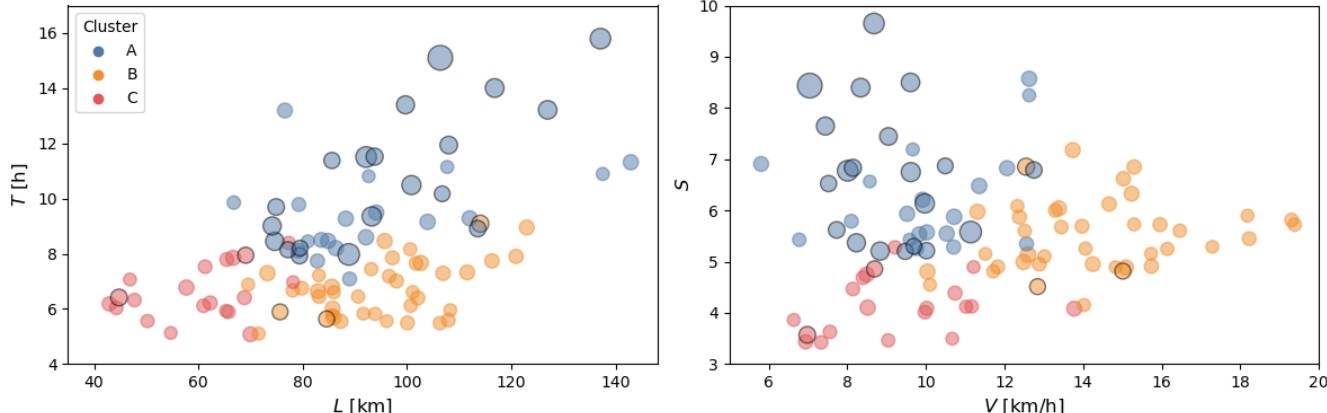

**Figure 6.** Scatter plot of (left) characteristic length $L$ and characteristic time $T$ (Eq. 9), and (right) characteristic speed $V$ and characteristic scale $S$ (Eq. 10). Shown are the top 100 precipitation events identified in Sect. 3. The point size indicates total precipitation volume similar to Fig. 2. The top 25 events are highlighted with black edge color. The point color indicates the cluster membership obtained from K-means clustering, which is applied to the normalized wavelet spectra $e^{j_{XY}, j_T} / \sum e^{j_{XY}, j_T}$.

In Fig. 6 (left), we show the characteristic length and characteristic time of each event, with colors indicating the assigned cluster. The three-cluster solution differentiates events as follows: (A) large spatiotemporal scale and slowly moving, (B) large spatiotemporal scale and fast moving, and (C) small spatio-temporal scale. This analysis suggests that spatio-temporal precipitation extremes are primarily governed by two features: Their characteristic speed $V$ and a combined space–time magnitude, the characteristic scale $S$. Both quantities have been introduced in Sect. 2.2. In Fig. 6 (right), we display $S$ and $V$ for the top 100 events, clearly illustrating the differences between the three clusters with respect to these two variables. As already apparent in Fig. 5, high-volume precipitation events tend to propagate more slowly. This tendency is confirmed in Fig. 6, where we find that the top 25 events predominantly fall into Cluster A.

We further analyzed if there is a trend towards changes in the spatiotemporal characteristics with time in the data. We tested for linear trends in the previously introduced characteristic parameters ($L, T, V, S$), as well as in the total precipitation volume and the annual number of identified events. The analysis was conducted for both, the full time series and separately for each cluster. We fitted a linear regression using maximum likelihood estimation and performed a t-test on the slope. The results are summarized in Tab. 2. Further details of the statistical test and graphical illustrations are provided in appendix Sect. B. Overall, we do not find any trends that meet the significance threshold of 0.01, which is likely due to the relatively short length of the time series. We do find indications of a decrease in the characteristic velocity of small-scale events (cluster C), but with a p-value of exactly 0.05, this result cannot be considered statistically significant. A longer observation period will be necessary to obtain more conclusive evidence in this regard.




**Table 2.** Slope for the linear trend analysis illustrated in Fig. B1. p-value for slope $\neq 0$ in brackets (t-test).

|  | $L$ [km] | $T$ [h] | $V$ [km/h] | $S$ | Prec. Vol. [V] | N Events |
|---|---|---|---|---|---|---|
| Cluster A | -0.17 (p=0.73) | -0.03 (p=0.65) | -0.005 (p=0.92) | -0.015 (p=0.62) | 8.4 $10^8$ (p=0.68) | / |
| Cluster B | -0.30 (p=0.39) | -0.02 (p=0.42) | -0.007 (p=0.90) | -0.017 (p=0.34) | 3.4 $10^8$ (p=0.61) | / |
| Cluster C | -0.13 (p=0.73) | 0.07 (p=0.04) | -0.114 (p=0.05) | 0.015 (p=0.41) | 2.3 $10^8$ (p=0.80) | / |
| All | -0.35 (p=0.28) | -0.05 (p=0.14) | 0.016 (p=0.74) | -0.028 (p=0.15) | -4.6 $10^8$ (p=0.63) | 0.034 (p=0.47) |

## 5 Physical foundations

In a final step, we aim to quantify the characteristic properties using additional data sources. Specifically, we consider the mean wind speed and the convective timescale. The convective timescale ($\tau_c$; Done et al., 2006) quantifies the characteristic time over which atmospheric instability, expressed as CAPE, is removed by convective heating and thus serves as an indicator of convective activity. As mentioned above, larger (smaller) values of $\tau_c$ correspond to weaker (stronger) large-scale forcing. Importantly, both variables originate from reanalysis data (Sect. 3) and are not derived from precipitation fields. Instead, they serve as descriptors of the prevailing atmospheric conditions during each event.

In Fig. 7, we compare the characteristic speeds derived from the wavelet spectrum with mean wind speeds from ERA5 reanalysis data over Germany during the respective events. The range of characteristic speeds falls well within the typical variability of synoptic-scale surface wind velocities, but the correlation is near zero. We however observe strong correlations ($r > 0.6$) with the mean wind speeds of the higher levels between 300 hPa and 700 hPa. The maximum correlations ($r = 0.68$) are at 500 hPa, indicating a dynamical linkage between the propagation of the observed precipitation systems and the mid-tropospheric flow. This finding is consistent with the well-established association between midlatitude cyclone activity, frontal passages, and precipitation documented in synoptic meteorology (e.g., Bott, 2016; Hofstätter et al., 2018). As shown in Appendix Fig. C1, when precipitation events are separated into their K-means clusters identified in Sect. 4.2, using the two independent variables $\tau_c$ and $v_{500hPa}$, the classification still performs remarkably well. We thus conclude that the characteristic speeds and scales derived from our method are physically meaningful and align with expected atmospheric dynamics.

Figure 8 presents the Pearson correlation matrix of all characteristic quantities introduced in this study - the characteristic scale $S$, speed $V$, length $L$, and time $T$ - together with the total precipitation volume, the convective timescale $\tau_c$, and the mean wind speed at 500 hPa. The latter level was selected because it showed the strongest correlation in the previous analysis compared to 1000, 750, and 250 hPa.

The correlations among the characteristic quantities summarize the findings from the previous sections. As expected, $L$ and $T$ show strong positive correlations with $S$, while $S$, $T$, and $V$ are mutually related, reflecting the internal consistency of these measures. In contrast, $S$ and $V$ are almost uncorrelated, confirming that the characteristic scale and speed represent independent aspects of precipitation system behavior.

The total precipitation volume correlates positively with $S$ and negatively with $V$, consistent with the results in sec 4.2, indicating that slowly propagating, large-scale systems tend to produce greater accumulated precipitation totals. We also find



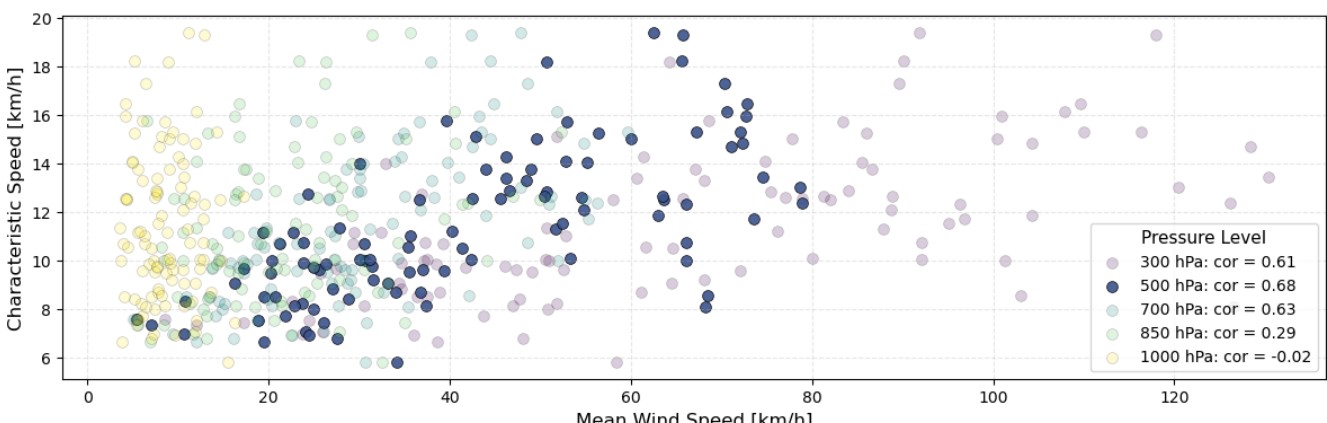

**Figure 7.** Scatter plot of the characteristic speed derived from the WT vs. the mean wind speed from ERA5 reanalysis at pressure levels of 300, 500, 700, 850, and 1000 hPa. Shown are the top 100 precipitation events identified in Sect. 3. Pearson correlations between characteristic and mean wind speeds are shown in the colorbar labels.

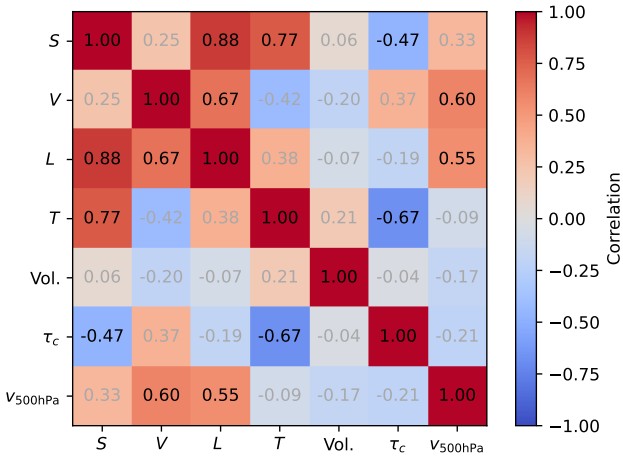

**Figure 8.** Pearson correlations between WT-derived characteristics $(S, V, L, T)$, precipitation volume, convective timescale $\tau_c$ and mean wind speed in 500 hPa. Grey numbers indicate non-significant correlations at p = 0.05 assessed with a permutation test (1000 permutations).

positive correlations between characteristic speed $V$ and mean wind speed at 500 hPa, as shown previously, linking precipitation system propagation to mid-tropospheric flow.

Finally, the convective timescale $\tau_c$ correlates negatively with the $S$, suggesting that short $\tau_c$ - that is, strong large-scale forcing - tends to promote more extensive, large-scale precipitation systems, whereas longer $\tau_c$ values favor smaller, more localized convective systems. Note that $\tau_c$ represents the atmospheric adjustment or CAPE consumption timescale, while $T$ denotes the duration of the precipitation system itself; hence, the two are expected to be negatively correlated.



## 6   Summary & Conclusions

We investigate the spatiotemporal characteristics of extreme precipitation events over Germany using WT. The WT is a spectral filtering method that decomposes the total energy of a signal into contributions from different scales or sizes. Our analysis considers three dimensions: two spatial and one temporal. However, we distinguish only between a combined spatial scale in the x- and y-directions and the temporal scale of precipitation events. Although a further separation of the two space dimensions is possible, a single spatial scale is consistent with the established scale-based view of the climate system (see, e.g., Kraus,
305 2004).

We identify precipitation events in the northern hemispheric summer months across Germany based on their total accumulated precipitation within a confined spatial and temporal window. This allows us to capture events with a strong potential for hydrological and societal impacts, for example when they occur over river catchments or in areas with limited runoff capacity.

In fact, the top six events that we identify include several well-documented cases previously studied in the literature (see
Ulbrich et al., 2003; Ludwig et al., 2023), which confirms that our selection criterion is appropriate for identifying high-impact precipitation events. They also span a broad range of spatial and temporal scales, offering a solid basis for demonstrating the capabilities of our method. Although we initially rely on the simplified concept illustrated by Kraus (2004), we demonstrate in Sect. 5 that our results are physically plausible.

To move from a detailed analysis of individual events to a broader examination of the top 100 events in Sect. 4.2, we
summarize the mean spatiotemporal characteristics of each event. Again following the simplified concept illustrated by Kraus (2004), we provide a characteristic length $L$, a characteristic time $T$, a characteristic speed $V$ and a combined spatiotemporal magnitude, which we denote as the characteristic scale $S$ for each event. Using the top six events as an example, we demonstrate the suitability of this approach.

Extending our analysis to the top 100 precipitation events, we apply statistical clustering to the wavelet spectra. In doing so,
we aim to identify distinct patterns and structures in the spatiotemporal characteristics of precipitation extremes. Our analysis reveals two key parameters: the characteristic scale, and the characteristic speed. By introducing the characteristic scale, we identify two independent characteristics: speed and scale. This separation enables a more straightforward interpretation of the spatiotemporal characteristics of precipitation events.

Most of the 25 strongest events, measured by their total precipitation within a confined area and time period, are charac-
terized by relatively slow characteristic speeds and large characteristic scales. This relationship becomes even clearer when considering the correlations between these variables. We therefore conclude that the spatiotemporal characteristics of precipitation events are of fundamental importance. Approaches such as the one proposed in this study can substantially advance our understanding of precipitation processes and their response to climate change. While we do not find significant changes in these characteristics over the observational period - likely due to the limited length of the available time series - studies based
on climate model simulations already indicate potential changes in extreme precipitation events under future climate conditions (see e.g. Hundhausen et al., 2024).





In a final step, we correlate the characteristic variables of precipitation events with variables describing atmospheric dynamics. We find that the characteristic speed is correlated with the mean wind speed at 500 hPa. Furthermore, by means of the convective time scale, we link large-scale precipitation events to large-scale atmospheric processes. These results confirm the

physical plausibility of our analysis and highlight its suitability for establishing quantitative links between precipitation events and atmospheric dynamics. A more comprehensive analysis, incorporating convection-permitting climate model simulations, is planned for future work.

*Data availability.* The RadKlim gauge-adjusted radar dataset, one-hour precipitation sums (RW) is provided by the German Weather Service (Winterrath et al., 2018). ERA5 output data were accessed through the XCES (ClimXtreme Central Evaluation System) at the Deutsches

Klimarechenzentrum (DKRZ).

## Appendix A: Assessment of K-means Cluster Size

To evaluate the quality and robustness of the K-means clustering results, we adopt a twofold validation approach. Cluster quality is assessed using the silhouette score (cf. rou, 1987), while stability under sampling variability is evaluated via the adjusted Rand index (ARI, Hubert and Arabie 1985).

Figure A1 shows the silhouette scores for cluster sizes ranging from 2 to 10. The silhouette score measures how well each data point fits within its assigned cluster compared to other clusters, with values ranging from $-1$ (poor fit) to 1 (clear separation). We find a maximum score of 0.31 for a three-cluster solution, indicating a moderate but meaningful cluster structure. Based on this result, we proceed with three clusters in the subsequent analysis. To evaluate the robustness of this three-cluster solution, we generate 1000 random subsamples, each containing 80% of the top 100 events (sampled without replacement).

For each subsample, we compute both the ARI and silhouette score. The distribution of these scores is shown as box plots in Fig. A1. The silhouette scores show a stable distribution centered around the observed value, with only a few isolated outliers (median = 0.30). The ARI further indicates a high degree of structural stability across subsamples, with a median value of 0.80. Taken together, these results suggest that the clustering solution provides both consistent inter-cluster separation and robust cluster assignment with respect to sampling variability.




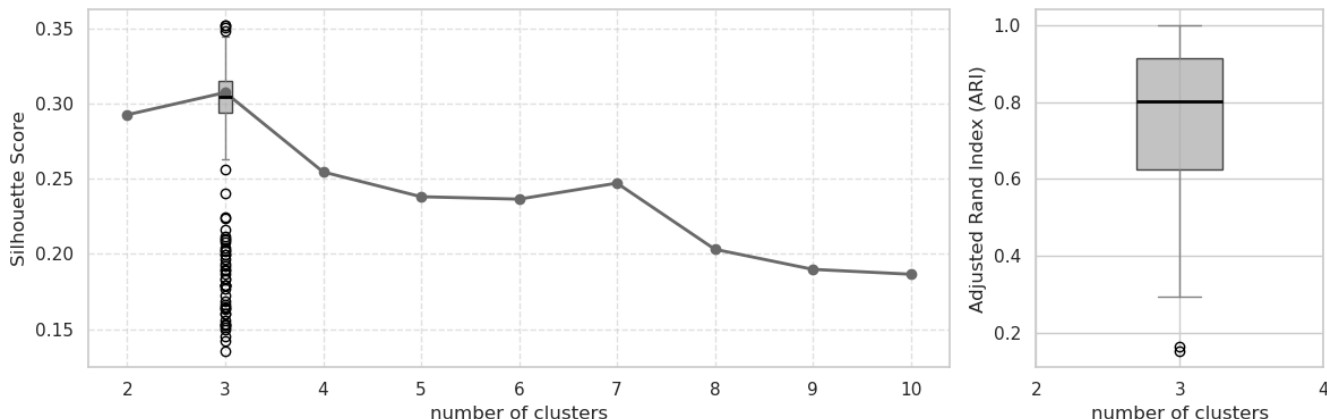

**Figure A1.** Silhouette score (left) and Adjusted Rand Index (ARI, right) for K-means clustering of the wavelet spectra of the top 100 precipitation events identified in Sect. 3. The silhouette score is shown for cluster sizes ranging from $k = 2, \ldots, k = 10$. The boxplots are based on a fixed number of three clusters and represent the distribution of silhouette scores and ARI values calculated from 1000 random subsamples (each comprising 80% of original data).





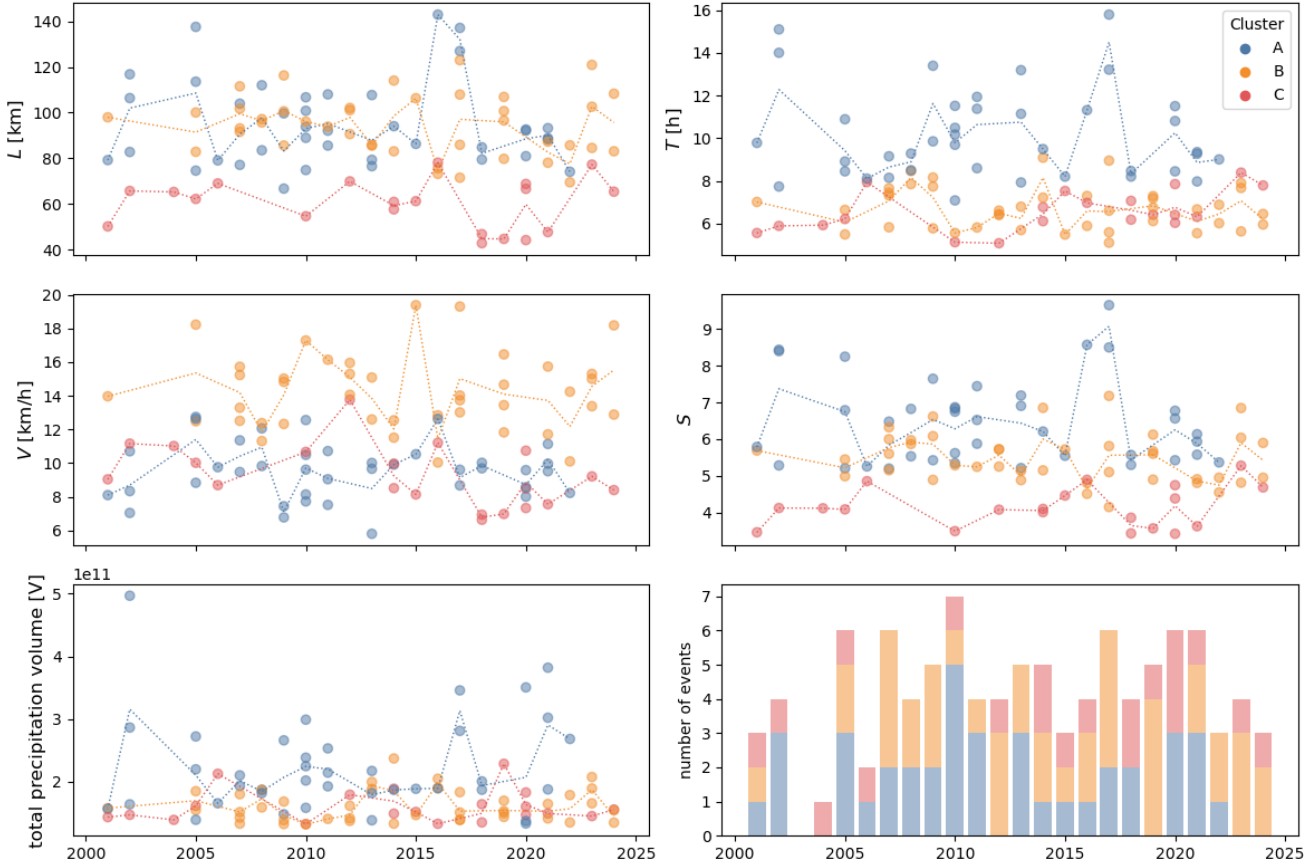

**Figure B1.** Scatter plots of the characteristic length $L$, characteristic time $T$, characteristic speed $V$, characteristic scale $S$, total precipitation volume (similar to Fig. 2) and number of events for each year. Shown are the top 100 precipitation events identified in Sect. 3. The points are colored according to cluster assignments obtained via K-means clustering similar to Fig. 6. Dotted lines indicate average values for each year and cluster.

## Appendix B: Testing for changes through time

In this paper, we analyse precipitation data from a total of 24 years. This naturally raises the question of whether we can find evidence of changes over time. In Sect. 4.2 we therefore present the results of a statistical testing of linear trends. We examine the key characteristics as introduced in this paper, namely length, time, speed, scale, total precipitation volume and, additionally number of events per year within the top 100. Figure B1 illustrates these six variables with time. The significance testing was performed using the integrated Wald test in the **sciPy** Python function *lineregress*, whose null hypothesis is that the slope is zero.




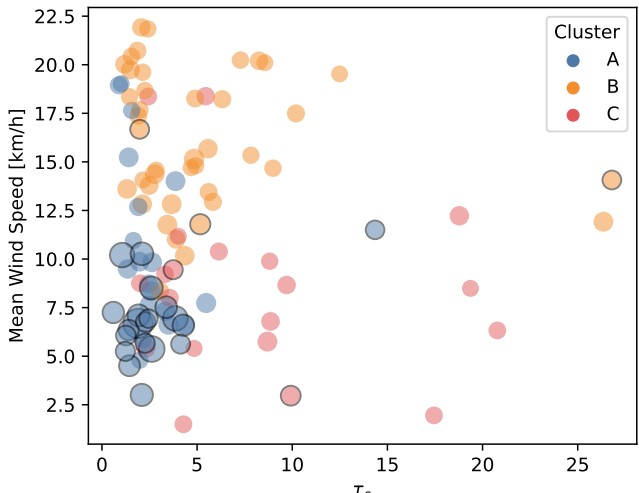

**Figure C1.** Same as Fig. 6 but for convective timescale $\tau_c$ and mean wind speeds in 500hPa.

## Appendix C: Connections with atmospheric dynamics

*Author contributions.* **Svenja Szemkus**: Conceptualization; Formal analysis; Visualization; Writing (original draft preparation).

**Sebastian Buschow**: Conceptualization; Formal analysis; Writing (review and editing).

**Petra Friederichs**: Conceptualization; supervision; Funding acquisition; Writing (review and editing).

*Competing interests.* The authors declare that they have no conflict of interest.

*Acknowledgements.* This work was conducted as part of the ClimXtreme II - Module B project funded by the Bundesministerium für Bildung und Forschung (BMBF FKZ 01LP2323A). We are grateful to our project partners Marco Oesting and Carolin Forster from the University of Stuttgart for valuable suggestions and fruitful discussions. This work used resources of the Deutsches Klimarechenzentrum (DKRZ) granted

by its Scientific Steering Committee (WLA) under project ID bm1159.



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
