# Peer review of "Revealing the structure of precipitation extremes: a spatio-temporal wavelet approach"

_EGUsphere, 2025_

## Referee Comment (RC1)

**Summary:** The paper presents a wavelet-based analysis of the spatiotemporal characteristics of extreme summer precipitation events in Germany using high-resolution radar observations. By deriving characteristic length, time, scale, and speed from a combined spatial-temporal wavelet framework, it captures precipitation extremes beyond traditional duration- or area-based metrics. While the approach is methodologically robust and well justified, a clearer articulation of the research gap, uncertainties, and physical interpretation of some results would strengthen the manuscript. Overall, the study provides a valuable framework for understanding and comparing precipitation extremes.

**Abstract**

1. Briefly stating the specific research gap (what is currently unknown about the spatio-temporal organisation of extremes in Germany) would strengthen the abstract.
2. It would be helpful to clarify whether the novelty lies in the metrics (characteristic time, length, and speed) themselves or in their application to radar-based extremes over Germany.
3. The results are currently described in general terms. It would be useful to add one concrete or quantitative example to make 'systematic patterns' more tangible for the reader.
4. Including a brief note on uncertainties or limitations associated with radar data or the wavelet decomposition approach would provide a more balanced view.

**Introduction**

1. The Introduction provides a strong motivation, but it would be helpful to explicitly state the research gap early on- what is unknown about the spatio-temporal structure of extreme precipitation in Germany that this study addresses?
2. Lines 61-70- Including a brief note on uncertainties associated with radar data or the wavelet decomposition approach would be helpful.
3. Lines 65-69- a short remark on how the metrics (e.g., characteristic scale and characteristic speed) could be useful beyond this study (e.g., for model evaluation or future climate projections) would broaden the perceived relevance.

**Theory**
1. Line 85- A short clarification of how alternative wavelet families were evaluated would improve transparency.
2. Lines 120-127- It would be beneficial to highlight the advantage of combining 1D temporal and 2D spatial WTs over a fully 3D WT.

**Data & Preprocessing**

1. Line 155- A short statement clarifying why these specific pressure levels were chosen in ERA5 would strengthen the physical interpretation.
2. Line 166- It may be useful to briefly discuss how the 72-hour separation criterion affects the number and independence of events.

3. Line 181- Acknowledging potential implications for events extending beyond the RadKlim spatial domain could strengthen the discussion.

**Analysis**

1. Section 4.1
    a. Lines 191-193- While the 2002 and 2021 events are noted among the top five, briefly clarifying if they represent different precipitation regimes (multi-day vs. short, intense convective events) would strengthen the case study motivation.
    b. Lines 200-203 note that 1D and 2D WTs treat space and time separately. A brief remark on how this limitation might affect interpretation or potential biases in estimating characteristic scales could benefit the readers.
    c. The comparison across the top six events is informative. Commenting on how representative these events are of the full sample would help contextualise the results.
        - Minor- there is a misalignment in the Table reference (mentioned Tab. 2) on Line 226 and the Table caption on Line 233 (mentioned Table 1).
2. Section 4.2
    a. It would be helpful to briefly justify the choice of K-means clustering over other clustering methods.
    b. A short physical interpretation of each cluster (in the three-cluster solution) in terms of precipitation-generating processes would be helpful.
    c. The observed relationship between precipitation volume and slower propagation is interesting; clarifying whether this is statistically tested or primarily descriptive would add clarity.
    d. In the trend analysis, it would benefit the readers by a brief discussion on the implications of limited sample size and short record length would help contextualise the non-significant results.

**Physical foundations**

1. The use of independent reanalysis-based variables to assess physical foundations is well motivated; consider briefly noting why they complement the WT-derived metrics to reinforce the added value of incorporating reanalysis data.
2. A short discussion of why near-surface winds show lower correlation, and what would be the potential reason(s) for 500 hPa showing the highest correlation, could help readers interpret the physical meaning more easily.
3. Minor- Line 290- Correlation summary is useful. Consider briefly noting variability or exceptions across events (e.g., cases where slowly propagating systems do not produce the largest volumes) to provide a balanced view.

**Summary and Conclusion**
1. Minor, but conclusions should be in the past tense, e.g., line 300- it should be we 'investigated', line 302- 'considered', 'distinguished', line 306- 'identified', etc.

2. Overall, currently, the conclusions read more like a brief list of the rest of the manuscript. Instead, it would be helpful to have a synthesis of key findings in a broader context, interpreting their meaning, highlighting their significance, and outlining potential directions for future work.
3. Line 321- briefly clarifying why characteristic scale and speed emerge as dominant would strengthen the interpretation.
4. Lines 324-329- Including a short statement on uncertainty or variability around the observed association between high-impact events and slow-moving, large-scale systems would provide balance.

**Technical comments**

1. It is observed that 'see' or 'e.g.' is used in most of the citations throughout the manuscript. It would be useful to revise and remove it wherever necessary.
2. Appendix A: Lines 349-351: The resampling strategy (80% subsamples, without replacement) is well described; stating why 80% was chosen would improve reproducibility.
3. Appendix C: No text, apart from the section title, has been added to support Fig. C1. It currently reads incomplete.

---

## Referee Comment (RC2)

Review of the manuscript titled "Revealing the structure of precipitation extremes: a spatio-temporal wavelet approach" by Svenja Szemkus, Sebastian Buschow, and Petra Friederichs.

The authors present a new method to analyse three-dimensional spatio-temporal precipitation characteristics by extending an existing two-dimensional method. They apply this method to a dataset of severe precipitation in Germany and evaluate derived characteristics.

The submitted manuscript presents a reusable idea, which has the potential to change the way how precipitation fields are evaluated. However, the presentation could be made more clear to the reader. A more clear separation of theory and experiment will allow readers to adopt the proposed technique more easily. Therefore, I recommend to address some items before proceeding to publication.

Please see my general questions as questions that potential readers could have. Feel free to modify the text wherever you see fit.

**Minor comments**

General:

- Can the methodology be applied to the whole dataset? Is it a necessary precondition to select individual events, as you did?

- What is the purpose of finding the 50x50 km event window, when the wavelet transform is applied to the whole domain?

- Why didn't you fill the whole 128 elements in the time dimension with data, but 94 elements and filled the rest with zeroes?

- Why didn't you apply the WT continuously for 3-day and compare the evolution over, e.g., two weeks? Is the result jumpy?

- What is the benefit of the proposed method? What do the results reveal in terms of 'structure'?

Specific comments:

- L25: What is "high-quality"? Rewrite?

- L82-84: Briefly state why you choose DT-CWT over, e.g., Daubechies. Why would you choose more complex, if you find no substantial performance?

- L60-64: This is a crucial jump. First you introduce the method, then you say that there is some continuous data, and then you select 100 events. I think there is a gap to be filled. It would be helpful to articulate on what data the method can be applied to, e.g., only extreme cases, or also to a whole reanalysis?
  Are you applying the method on the whole data of 2001-2024, or 100 events?
  At this point I am confused if the selection of events is necessary? Or can I apply it on 30-years of reanalysis too? Why 100 events and not the whole dataset?
  Maybe you put details in section 2 (next comment)

- Similarly as above, L145-146: The transition from section 2 to 3 is rough.
  At the end of section 2, the reader has the method, but doesn't necessarily know on which kind of data it can be applied, etc.
  Provide details to:
  1) What are the limitations of the method?
  2) What are there requirements for the data/observations? can it be applied to satellite-derived precipitation data and station data?
  3) How long needs the timeseries to be?
  Finally, a reader would benefit from a step-by-step list of what a user needs to do to apply the method to a new dataset.

- L153: I doubt that 24 years of data are suitable for long-term climate trend analysis?!

- Title: "Revealing the structure of precipitation extremes":

    o From the description of the method I concluded that it gives characteristics not only for extremes but also the whole distribution, right?

    o Are the results characteristics of extremes or of the whole German domain during the 94 hours that went into the wavelet transform?

- L217: " the orientation along a straight line can be attributed to the movement of precipitation systems through space." Please clarify. Also, the reference is German, so please briefly describe what you mean.

- L302-305: This can be shortened and formulated more boldly, e.g. remove "only" and the sentence "Although ..." could be placed in the methods section.

- L312: " Although we initially rely on the simplified concept illustrated by Kraus (2004)" This half-sentence is confusing here. Maybe it can be (re)moved?

Typos:

- L48: "its"

- L226: "Tab. 2" You meant table 1?